# Diet choice: The two-factor host acceptance system of silkworm larvae

Kana Tsuneto[1], Haruka Endo[1,2]*, Fumika Shii[1], Ken Sasaki[3], Shinji Nagata[2], Ryoichi Sato[1]*

1 Graduate School of Bio-Applications and Systems Engineering, Tokyo University of Agriculture and Technology, Koganei, Tokyo, Japan, 2 Department of Integrated Bioscience, Graduate School of Frontier Sciences, The University of Tokyo, Kashiwa, Japan, 3 Graduate School of Agriculture, Tamagawa University, Machida, Tokyo, Japan

* haruka@edu.k.u-tokyo.ac.jp (HE); ryoichi@cc.tuat.ac.jp (RS)

## Abstract

Many herbivorous insects are mono- or oligophagous, having evolved to select a limited range of host plants. They specifically identify host-plant leaves using their keen sense of taste. Plant secondary metabolites and sugars are thought to be key chemical cues that enable insects to identify host plants and evaluate their quality as food. However, the neuronal and behavioral mechanisms of host-plant recognition are poorly understood. Here, we report a two-factor host acceptance system in larvae of the silkworm *Bombyx mori*, a specialist on several mulberry species. The first step is controlled by a chemosensory organ, the maxillary palp (MP). During palpation at the leaf edge, the MP detects trace amounts of leaf-surface compounds, which enables host-plant recognition without biting. Chemosensory neurons in the MP are tuned with ultrahigh sensitivity (thresholds of attomolar to femtomolar) to chlorogenic acid (CGA), quercetin glycosides, and β-sitosterol (βsito). Only if these 3 compounds are detected does the larva make a test bite, which is evaluated in the second step. Low-sensitivity neurons in another chemosensory organ, the maxillary galea (MG), mainly detect sucrose in the leaf sap exuded by test biting, allowing larvae to accept the leaf and proceed to persistent biting (feeding). The two-factor host acceptance system reported here may commonly underlie stereotyped feeding behavior in many phytophagous insects and determine their feeding habits.

## Introduction

Diet choice is essential for survival in all organisms. Many herbivorous insects have adapted to a limited range of host plants by overcoming plant defenses against insect herbivory, resulting in mono- and oligophagous insects (so-called specialist insects) [1]. In parallel, specialists are able to precisely distinguish their host plants from other nonhost plants in their ecosystem using chemical senses, including gustation and olfaction [2]. In general, gustation is important in determining the acceptance or rejection of a potential food source, whereas olfaction is required when searching for host plants from a distance [3].

**Data Availability Statement:** All relevant data are within the paper and its Supporting Information files.

**Funding:** 1. Japan Society for the Promotion of Science (JSPS) KAKENHI Grant Number 17K19261

to RS (https://kaken.nii.ac.jp/ja/grant/KAKENHI-PROJECT-17K19261/) Grant Number 18J00733 to HE (https://kaken.nii.ac.jp/ja/grant/KAKENHI-PROJECT-18J00733/) 2. The funders had no role in study design, data collection and analysis, decision to publish, or preparation of the manuscript.

**Competing interests:** The authors have declared that no competing interests exist.

**Abbreviations:** AN, antenna; ANOVA, analysis of variance; CGA, chlorogenic acid; ISQ, isoquercitrin; LS, lateral styloconic sensillum; MG, maxillary galea; MP, maxillary palp; MS, medial styloconic sensillum; Q3R, quercetin-3-$O$-rhamnoside; βCD, methyl-β-cyclodextrin; βsito, β-sitosterol.

Plant-feeding insect larvae exhibit a typical stepwise feeding behavior, comprising foraging, palpation, biting, and swallowing [4,5]. Sensory exploration by palpation of the leaf surface is important for host-plant selection in various plant-feeding insects, such as caterpillars, beetles, and grasshoppers [6–9]. For example, the meadow grasshopper *Chorthippus parallelus* sometimes rejects a diet following palpation using peripheral chemosensory organs without biting [6]. Conversely, leaf-surface extracts of the host plant *Poa annua* induced biting behavior in the migratory locust *Locusta migratoria*, whereas those of nonhost plants were rejected [10]. Thus, insects use leaf-surface compounds as chemical cues for host-plant selection, but the neuronal basis for this behavior and to what extent the leaf-surface behavior contributes to whole host-plant selection or leaf choice are poorly understood.

Feeding in insect herbivores is controlled by their physiological responses to phytochemicals that determine the acceptance or rejection of a leaf based on the balance of feeding stimulants and deterrents contained in plant leaves [3,11,12]. Chemical ecological studies have identified plant secondary metabolites as feeding stimulants and deterrents for many insect herbivores [2]. For example, with regard to the silkworm *Bombyx mori*, a specialist on mulberry leaves, Hamamura and his colleagues identified nonvolatile compounds from mulberry leaves as key signals: isoquercitrin (ISQ) and β-sitosterol (βsito) as biting factors and chlorogenic acid (CGA) as a feeding stimulant that increases feeding amounts in a long-term assay [11,13,14,15]. Thus, feeding stimulants and deterrents in the diet choice have been evaluated by their effects on the number of bites and feeding amounts in most cases. However, these observations do not always reflect the instantaneous behavior of insects that accept or reject certain plants [16], and thus, careful investigation of behavior at the leaf surface is required for further understanding of role of plant secondary metabolites in host-plant selection.

In the present study, we assigned chemical host cues to taste organs and feeding behaviors to precisely understand the mechanisms underpinning host acceptance in the silkworm. We found that ultrasensitive chemosensory neurons in the maxillary palp (MP) of silkworm larvae detect a set of key mulberry compounds, including CGA, quercetin glycosides, and βsito, inducing a test bite. Sucrose and *myo*-inositol in leaf saps stimulate lateral sensilla in the maxillary galea (MG) to induce continuous biting and acceptance of the leaf. We propose a two-factor host acceptance system, controlled by 2 peripheral chemosensory organs and mainly driven by 6 phytochemicals, that underlies oligophagy in silkworms.

## Results

### Two peripheral chemosensory organs control two-step biting behaviors in the silkworm

To clarify the mechanism of host acceptance by silkworm larvae, we observed larval feeding of a host leaf from white mulberry *Morus alba*. When a silkworm encounters a leaf, it first palpates the leaf edge using a peripheral chemosensory organ known as the maxilla, intermittently bites the edge several times, and finally engages in continuous biting (2–3 times per second) with its head shaking in the dorsoventral direction along the leaf edge (Fig 1A; S1 Movie). The intermittent biting with palpation and the continuous biting with head-moving are termed test biting and persistent biting [4,5]. We hypothesized that sensing of chemical cues from an *M. alba* leaf via the maxilla induces test biting because test biting always occurs after palpation with the maxilla. The maxilla consists of the MP and MG (Fig 1B). To assess the roles of the MP and MG in the induction of test biting, we used MP- or MG-ablated larvae. MP-ablated larvae showed palpation, but no test or persistent biting (Fig 1C and 1D; S2 Movie). MG-ablated larvae showed palpation and test biting, stopped biting within 1 minute, and did not progress to persistent biting (Fig 1C and 1D; S3 Movie). When we ablated an olfactory organ

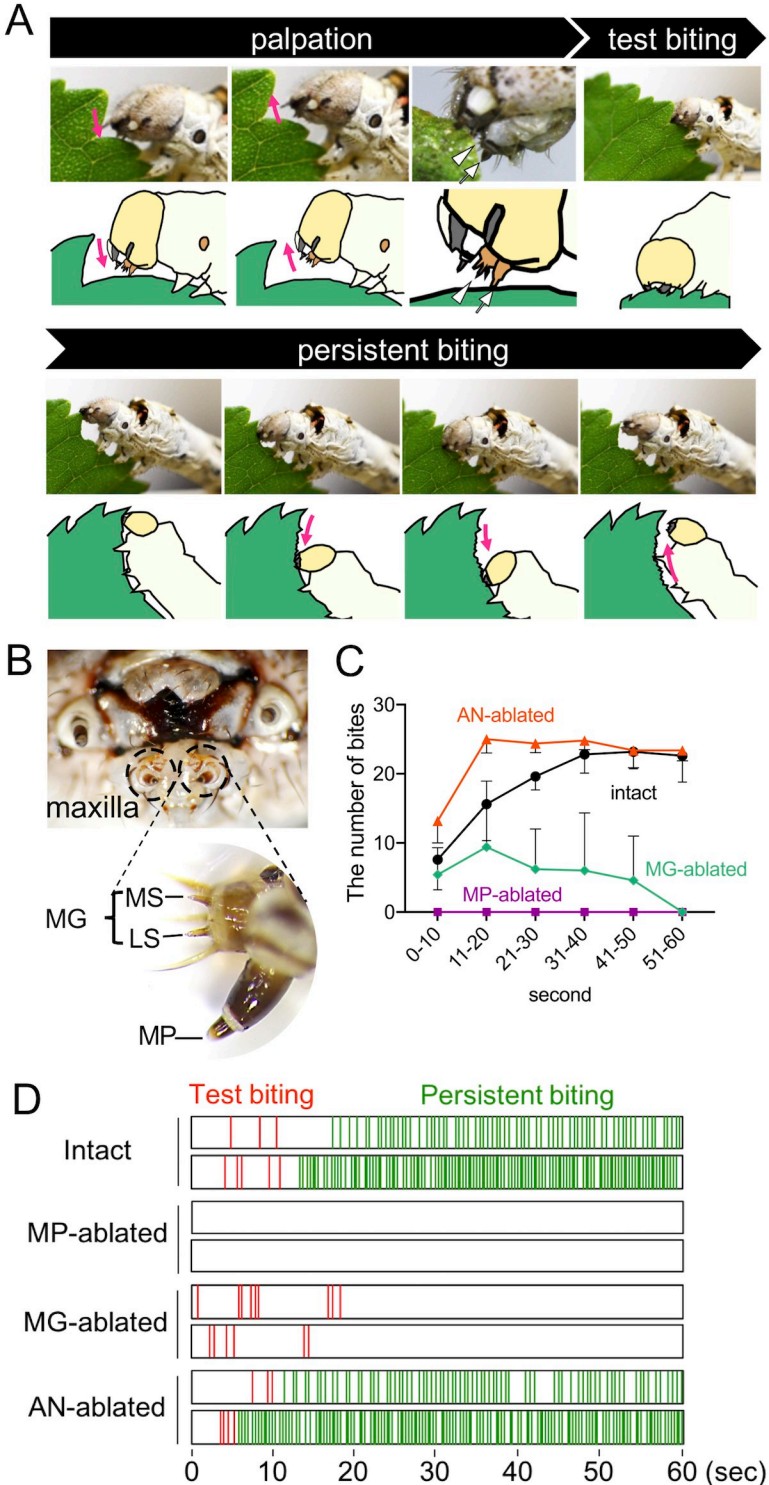

**Fig 1. MP and MG are responsible for test and persistent biting.** (A) Feeding on mulberry leaves by silkworm larvae.
(1) Palpation: a silkworm larva first palpates the leaf surface with its maxilla (MP and MG) for 5–30 seconds. The white
arrow and arrowhead indicate MP and MG. (2) Test biting: the larva bites the leaf edge several times intermittently
during palpation. (3) Persistent biting: the larva nibbles the leaf edge repeatedly (2–3 times per second) with its head
moving in the dorsoventral direction along the leaf edge. Magenta arrows indicate the direction of head movement. (B)
Mouthparts of a silkworm larva. Upper larval mouthparts including 1 pair of maxilla. Lower higher-magnification view
of the maxilla. The maxilla consists of the MG and MP. The MG has 2 gustatory sensilla, the LS and MS. (C) Frequency

of biting by intact, MP-ablated, MG-ablated, and AN-ablated larvae of mulberry leaves over a 10-second period. Data are means ± SD (*n* = 5). For numerical raw data, please see S2 Data. (D) Representative raster plots of the timing and duration of the biting behavior of larvae in (C). Red and green lines indicate test and persistent bites, respectively. AN, antenna; LS, lateral styloconic sensillum; MG, maxillary galea; MP, maxillary palp; MS, medial styloconic sensillum.

antenna (AN), the larvae showed palpation, test biting, and persistent biting similar to intact larvae (Fig 1C and 1D; S4 Movie). Note that AN ablation increased the number of bites during first 30 seconds, although the reasons for this are unclear. These results indicate that the MP and MG are essential for the induction of test and persistent biting.

## MP recognizes edible leaves by sensing leaf-surface compounds and induces test biting

To assess how such MP- and MG-controlled biting behaviors contribute to host acceptance of the silkworm, we observed feeding behavior towards leaves of various plant species. The silkworm is a specialist for some *Morus* species, including *M. alba*. In addition, the leaves of several Cichorioideae plants of the Asteraceae are consumed in relatively small amounts by silkworm larvae [17,18]. Of the larvae, 88.9% ± 5.3% showed test biting of *M. alba* within 1 minute after reaching the leaf edge, compared with 70.0% ± 10.0% and 63.3% ± 13.3% for 2 Cichorioideae plants, *Sonchus oleraceus* and *Taraxacum officinale* (Fig 2A). Of the larvae, 80% ± 11.5%, 53.3% ± 14.5%, and 30% ± 10% proceeded to persistent biting of *M. alba*, *S. oleraceus*, and *T. officinale*, respectively. In contrast, the larvae had a lower probability of test biting (3%–33%) of 12 inedible leaves, which were finally rejected by most of them without persistent biting (Fig 2A and 2B). Thus, the probability of test biting was higher for edible than for inedible leaves, and persistent biting was induced only by edible leaves. These results suggest that silkworm larvae select their diet sequentially at 2 points, before test and persistent biting, and that host-plant recognition is largely completed when they make a first bite.

Insects are likely to sense leaf-surface compounds during palpation [16]. To elucidate whether leaf-surface compounds induce test biting, we wiped *M. alba* leaves with water or methanol, which markedly decreased the proportion of larvae showing test biting (Fig 2C). Conversely, 76.7% ± 8.8%, 37.5% ± 10.3%, and 36.0% ± 10.8% of the larvae showed test biting of filter paper treated with methanol and leaf-surface extracts of *M. alba*, *S. oleraceus*, and *T. officinale*, respectively (Fig 2D; S1A–S1C Fig). MP ablation diminished test biting towards filter paper treated with an extract of *M. alba* leaf surface (Fig 2D). Furthermore, tip recording of the sensilla in the MP [17] revealed that MP neurons responded to the leaf-surface extract (Fig 2E). These findings indicate that compounds in edible leaf-surface extracts stimulate the MP and trigger test biting.

## Test biting requires a set of host-plant compounds detected by ultrasensitive MP sensory neurons

To identify inducers of test biting, we searched for secondary metabolites in edible leaves of *M. alba*, *T. officinale*, and *Lactuca indica* [18,19] in the plant-metabolite database KNApSAcK [20] because secondary metabolites are thought to be key chemical cues for host-plant recognition [2]. The search yielded CGA and quercetin-3-*O*-rhamnoside (Q3R). CGA reportedly increases the amount of food intake by the silkworm [15]; Q3R is an analog of ISQ, which reportedly induces biting by the silkworm [14]. In addition, we focused on βsito because it also reportedly induces biting by the silkworm [13]. We first recorded the responses of the MP and MG towards these compounds. Surprisingly, MP responded to the 4 compounds at the atto-molar and femtomolar levels (Fig 3A). In contrast, the MG did not respond to these compounds even at higher concentrations (S2B and S2C Fig). Next, we assessed whether the 4

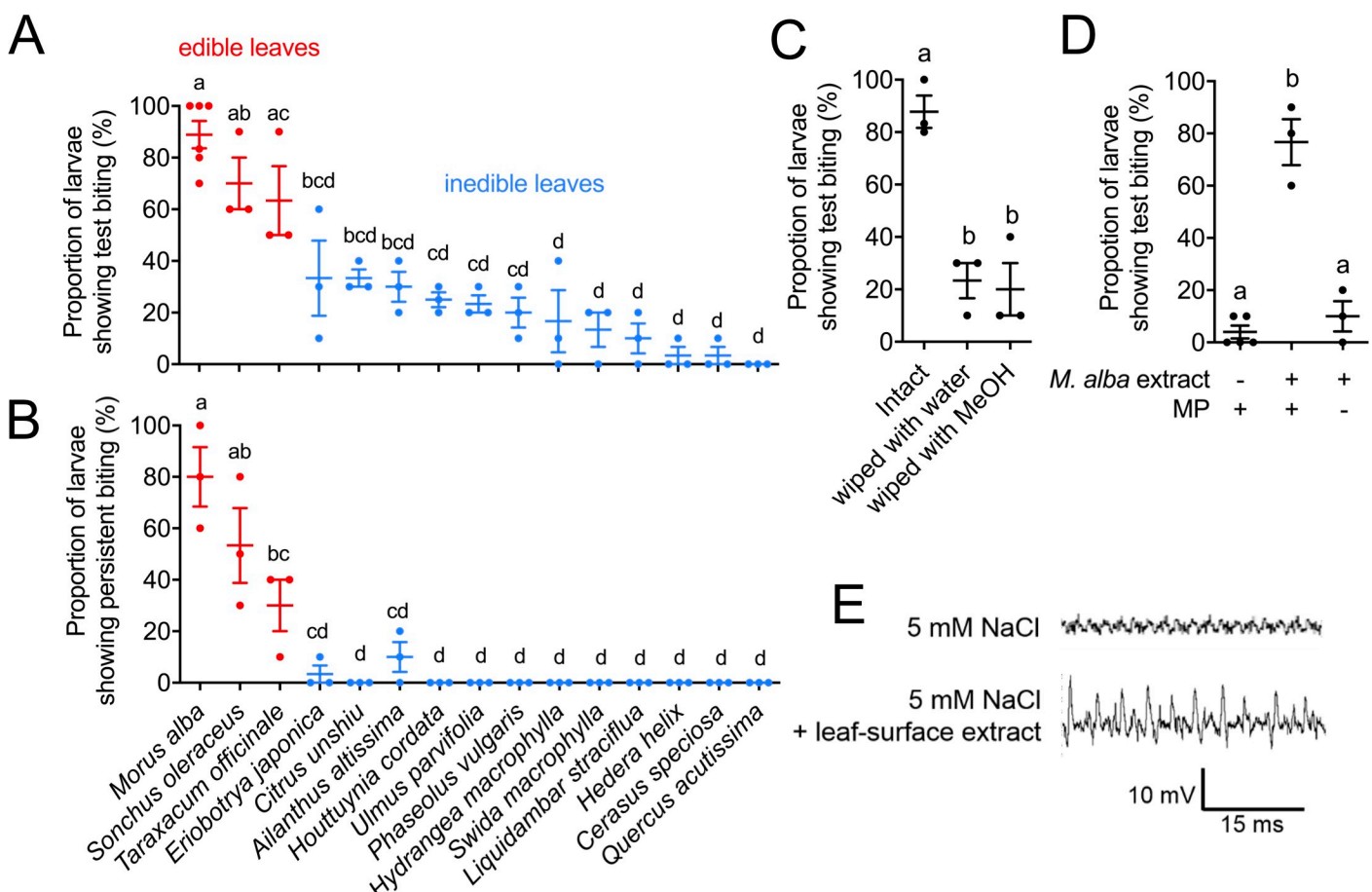

**Fig 2. The MP detects leaf-surface compounds and triggers test biting of edible leaves.** (A and B) Proportion of larvae showing test biting (A) and persistent biting (B) of edible and inedible leaves. (C and D) Proportion of larvae showing test biting of intact mulberry leaves or mulberry leaves wiped with sheets of Kimwipe (Nippon Paper Crecia Co., Tokyo, Japan) paper moistened with water or methanol (C) or filter paper treated with methanol or mulberry leaf-surface extract (D). MP + and MP− indicate intact and MP-ablated larvae. (A–D) Biting of 10 larvae was observed for 1 minute. Experiments were repeated as independent biological replicates ($n$ = 3–5). Data are means ± SE. The same letters indicate no significant difference ($P > 0.05$) by one-way ANOVA followed by Tukey post hoc test. (E) Typical spikes from chemosensory sensilla of the MP towards 5 mM NaCl (negative control) and leaf-surface extract (compounds from a leaf/10 mL). For numerical raw data, please see S2 Data. ANOVA, analysis of variance; MP, maxillary palp.

compounds induced test biting. Filter paper, which was treated with each single compound, mixtures of 2 compounds, and a mixture of ISQ, Q3R, and βsito induced test biting by 20%–40% of larvae. In contrast, mixtures of 3 compounds (CGA + ISQ + βsito and CGA + Q3R + βsito) and the mixture of all 4 compounds resulted in a high probability of test biting comparable to the *M. alba* leaf-surface extract (Fig 2D and Fig 3B) but did not induce persistent biting (S1D Fig). In agreement with the ultrasensitivity of the MP, filter papers treated with extremely dilute mixtures of CGA, ISQ, and βsito still induced test biting to some extent (Fig 3C). Meanwhile, a mixture of D-fructose, sucrose, D-glucose, and *myo*-inositol did not induce biting (Fig 3B). These results suggest that a trace amount of the set of CGA + ISQ/Q3R + βsito contribute to host recognition and induction of test biting.

### Sucrose and *myo*-inositol induce persistent biting via MG and modulate the amount of food intake

Next, we investigated the role of the MG in inducing persistent biting. The lateral styloconic sensillum (LS) in the MG of the silkworm larvae is involved in recognition of feeding

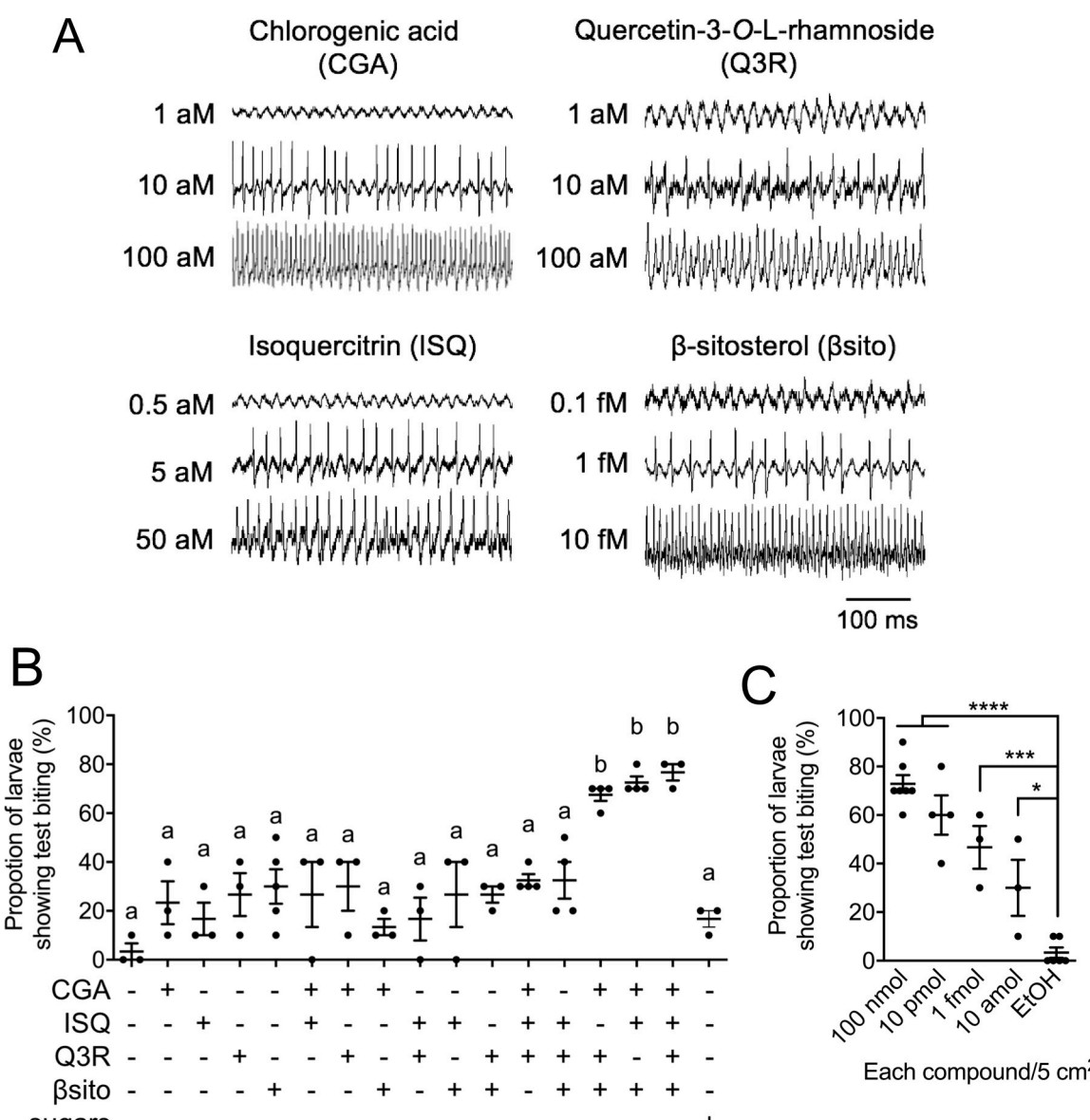

**Fig 3. A mixture of CGA, ISQ/Q3R, and βsito induces test biting by stimulating the MP.** (A) Typical electrophysiological recordings from the MP in response to CGA, ISQ, Q3R, and βsito. (B and C) Fraction of larvae showing test biting of filter paper treated with CGA, ISQ, Q3R, βsito, and sugars (sucrose, D-fructose, *myo*-inositol, and D-glucose) (B) and with an extremely dilute mixture of CGA, ISQ, and βsito (C). Filter paper was treated with 100 μL of 1 mM stimulant solution (100 nmol of each compound/5 cm$^2$) (B) or a dilution series of the CGA, ISQ, and βsito mixture (C). Biting by 10 larvae was observed for 1 minute. Experiments were repeated as independent biological replicates (*n* = 3–5). Data are means ± SE. Statistical analysis was performed using one-way ANOVA followed by Tukey post hoc test. The same letters indicate no significant difference (*P* > 0.05). An asterisk indicates a significant difference (*$P$ < 0.05; \*\*\*$P$ < 0.001; \*\*\*\*$P$ < 0.0001). For numerical raw data, please see S2 Data. ANOVA, analysis of variance; CGA, chlorogenic acid; ISQ, isoquercitrin; MP, maxillary palp; Q3R, quercetin-3-*O*-rhamnoside; βsito, β-sitosterol.

stimulants and has 3 neurons specifically tuned to D-glucose, sucrose, and *myo*-inositol, respectively, at around the millimolar level [21]. Therefore, we hypothesized that sugars in the leaf sap exuded by test biting stimulate the MG and induce persistent biting. Because feeding initiation is strictly regulated by test biting, we conducted an agar-based food intake assay using starved larvae [22] to simply evaluate persistent biting. In this assay, starved larvae no

longer exert specific preference for mulberry leaves and sometimes randomly bite; this biting substitutes for test biting, and consequently, persistent biting occurred in the absence of the inducers of test biting. Alternatively, unlike when feeding on leaves, the MG directly detected high concentrations of compounds at the agar surface during palpation, resulting in induction of persistent biting. Larvae fed an agar diet containing sucrose at >10 mM showed persistent biting (Fig 4A). The amount of food intake seemed to correlate with the duration of persistent biting. Indeed, a sucrose dose-dependent increase in the body weight was observed in intact and MP-ablated larvae (S3B and S3C Fig). The magnitude of the sucrose-induced increase in larval weight was significantly smaller in MG-ablated larvae than in intact larvae (Fig 4A), suggesting an important role of persistent biting via MG in modulating the amount of food intake. Meanwhile, *myo*-inositol and D-glucose themselves did not induce larval weight increase (Fig 4B; S3D Fig), whereas *myo*-inositol showed a supplemental effect in the presence of sucrose (Fig 4B) in agreement with a previous study [22]. These results suggest that sucrose and *myo*-inositol contribute to induction of persistent biting by stimulating MG.

Finally, we assessed whether the inducers of test biting (CGA, ISQ, and βsito) and sugars (sucrose, *myo*-inositol, and D-glucose) resulted in an increase in larval weight similar to that induced by *M. alba* leaf extract and intact leaf. A mixture of 10 mM sucrose, 5 mM *myo*-inositol, and 5 mM D-glucose, similar to the concentrations in *M. alba* leaves [23], resulted in an increase in larval weight in the presence, but not the absence, of a mixture of sugars, similar to an *M. alba* leaf and a leaf-extract–containing agar-based diet (Fig 4C). The test biting induced by CGA + ISQ + βsito accelerated the first bite (S4 Fig), which might cause persistent biting and consequently increase the total food intake. Therefore, we concluded that these 6 compounds are major phytochemical drivers of silkworm feeding (Fig 4D).

## Discussion

In the present study, we integrated chemical cues, chemosensory organs, and biting behaviors to provide new understanding of the mechanisms by which silkworm larvae accept mulberry leaves. Since the 1970s, entomologists have noticed that insects may identify their host plants at the leaf surface, but the mechanisms underlying this phenomenon remained largely unknown. We report here that ultrasensitive chemosensory neurons in the MP enable silkworm larvae to sense leaf-surface compounds and thus recognize their host plants. CGA, quercetin glycosides, and βsito were isolated as feeding stimulants for silkworm larvae, but it had been generally thought that these compounds could not explain the preference for mulberry leaves by silkworms because they are not specific to mulberry. Now, our findings suggest that the combination of the 3 compounds at the leaf surface explains the preference. Silkworm larvae do not feed on some Moraceae leaves, such as those of fig trees, but will feed on an agar-based diet containing both nonhost Moraceae leaves and mulberry leaves [18], suggesting that the nonhost Moraceae leaves do not contain feeding deterrents, but rather lack feeding stimulants. The composition of the inducers of test biting at the leaf surface can vary, even among the Moraceae. Alternatively, there may be limited plant species that produce the required combination within the habitat of *B. madarina*, an ancestor of the domesticated silkworm.

Host-plant recognition in larval feeding and adult oviposition seem to employ quite similar systems, driven by multiple host-plant compounds. Some butterflies are reported to require a specific combination of 6–10 host-plant compounds to induce egg laying [24–26]. This requirement is very similar to the host-plant acceptance mechanisms that we identified in silkworm larvae feeding. Furthermore, in the swallowtail butterfly *Papilio xuthus*, a specialist feeder on Rutaceae plants, a specific combination of the 3 neurons detecting multiple oviposition stimulants is responsible for oviposition [27]. The requirement of the combination of 3

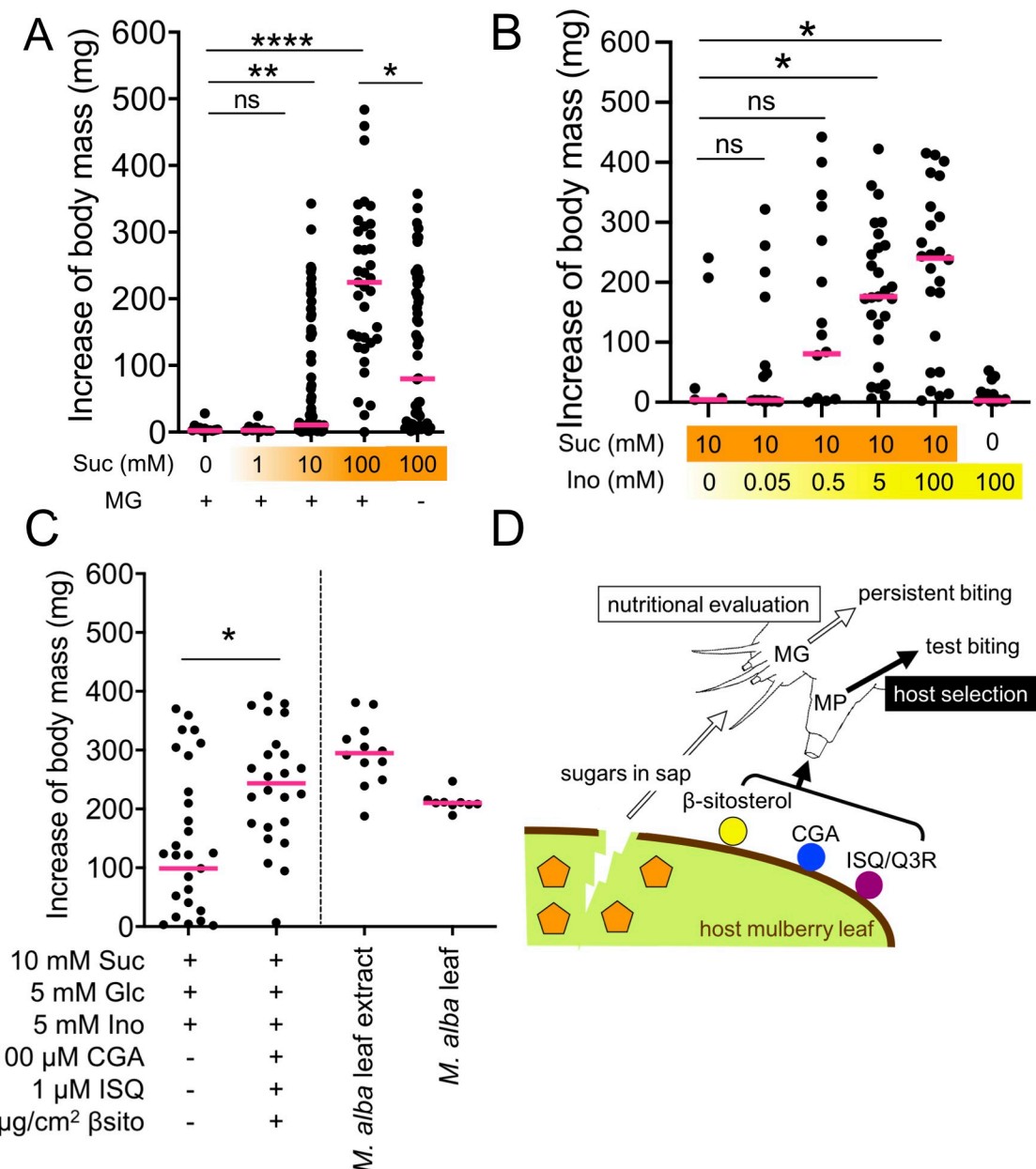

**Fig 4. Suc induces persistent biting by stimulating the MG in cooperation with Ino.** (A–C) Effects of Suc (A), Ino (B), and inducers of test biting CGA, ISQ, and βsito (C) on the increase in larval body mass by agar-based food intake assay. Fifth-instar larvae were starved for 3 days beginning at the end of molting. The body mass of the 3 larvae increased after feeding of the agar-based diet (9% cellulose and 1% agar) for 3 h. Magenta bars denote median. Statistical analysis was performed using Kruskal–Wallis test followed by Dunn test (A and B) and Mann–Whitney $U$-test (C). "ns" indicates no significant difference; an asterisk indicates a significant difference (*$P < 0.05$; **$P < 0.01$; ****$P < 0.0001$). MG+ and MG− indicate intact and MG-ablated larvae. (D) Proposed model of host recognition and acceptance by the silkworm. For numerical raw data, please see S2 Data. CGA, chlorogenic acid; Glc, D-glucose; Ino, *myo*-inositol; ISQ, isoquercitrin; MG, maxillary galea; MP, maxillary palp; Q3r, quercetin-3-*O*-rhamnoside; Suc, sucrose; βsito, β-sitosterol.

compounds to trigger test biting in silkworm larvae suggests that different chemosensory neurons are responsible for CGA, quercetin glycosides, and βsito and that the firing of multiple neurons is required to trigger test biting. Thus, insect herbivores may utilize their own combinations of phytochemicals for host-plant recognition in feeding and oviposition.

Our study clarifies an essential role of the MP in host-plant selection and feeding initiation in the silkworm. It was known that the MP has both gustatory and olfactory chemosensory neurons in lepidopteran insects [28], but its role in gustation had previously been poorly studied. We found that the silkworm MP is tuned to several mulberry compounds, but the specificities of more than 20 gustatory neurons in the MP [17] for chemical tastants such as sugars, amino acids, secondary metabolites, salt, and water remain largely uncharacterized. Future work will identify the neurons and receptor molecules that are essential for host-plant recognition. In addition, the possible role of olfaction by the MP in host-plant selection should be addressed. To our knowledge, the sensitivity of the attomolar and femtomolar ranges in gustation is the highest determined to date in insects and other animals. The leaf-surface cuticles are covered by a waxy layer, and therefore, the amounts of phytochemicals at the leaf surface are expected to be low. It is plausible that this ultrahigh sensitivity of the MP is necessary to detect trace amounts of inducers of test biting at the leaf surface.

The MG has been shown to play a main role in host-plant selection in lepidopteran insects [3,29]. The medial styloconic sensillum (MS) of the silkworm MG is known to respond to feeding deterrents, including coumarin, caffeine, and nicotine, at micromolar to millimolar levels [30,31]. Considering the two-factor host acceptance system, a probable role of the MS in the silkworm is to probe for toxic compounds in leaf sap generated by test biting. Most larvae exhibiting test biting on inedible leaves did not proceed to persistent biting, supporting the putative role of the MS in a final check to exclude inedible leaves by detecting feeding deterrents. Numerous information by previous studies on caterpillar neurophysiological responses of the MG to feeding deterrents [2,3,29] may suggest a similar role of MG in other lepidopteran insects. Note that the MG in some lepidopteran species respond to host-plant secondary metabolites [3,29], whereas silkworm MG did not respond to mulberry secondary metabolites we tested (CGA and quercetin glycosides) (S2B and S2C Fig). In these species, host-plant secondary compounds may stimulate the MG and induce persistent biting in combination with sucrose and *myo*-inositol. Meanwhile, insect herbivores often reject a leaf without biting. Glendinning and colleagues reported that *Manduca sexta* larvae rejected a diet treated with an extract from a nonhost plant by sensing it via the MP [32], suggesting the possibility that the MP senses feeding deterrents at the leaf surface with ultrahigh sensitivity. Thus, the two-factor host acceptance system likely comprises sensory responses to deterrents and stimulants by both the MP and MG.

By expanding existing models for caterpillars that suggest that feeding is governed by the balance of feeding stimulants and deterrents that stimulate the MG [3,33,34], the two-factor host acceptance system we proposed here may offer a general explanation of host-plant selection in lepidopteran insects. In monophagous/oligophagous caterpillars (i.e., specialists), host-plant recognition by MP is probably conserved, and the tuning of neurons in the MP to unique combinations of phytochemicals must reflect a history of insect adaptation to their host plants. In polyphagous caterpillars (i.e., generalists), feeding initiation may be less restricted by the MP; thus, rejection of foods may fall under control of responses by the MG to deterrent compounds as shown in previous studies [2,3]. The stepwise feeding behavior, comprising feeding initiation by recognizing specific foods and followed by sustained feeding, is reported in not only phytophagous insects like caterpillars [4,5] and grasshoppers [35] but a wider range of insects. In the fruit fly *Drosophila melanogaster*, the yeast feeding program starts with proboscis extension regulated by the labellum, and subsequent sustained feeding is regulated by taste pegs [36]. This suggests that the two-factor system for diet choice, utilizing different chemosensory organs and operating sequentially in time, is largely conserved beyond phytophagous insects. Further comparative studies using other insects will reveal a more concrete picture of the two-factor host acceptance system underpinning their feeding habits.

## Materials and methods

### Insects

The silkworm (*B. mori* Kinshu × Showa hybrid) eggs were purchased from Ueda Sanshu Ltd (Nagano, Japan). The silkworms were reared on an artificial diet, Silkmate 2M (Nihon-Nosan Co. Ltd., Kanagawa, Japan) with 16L-8D at 25°C. Larvae were provided with fresh diets every day to synchronize growth.

### Biting assay towards leaves and filter paper

Videos of silkworm feeding on leaves and filter paper and filter paper were taken by a D5100 camera (Nikon, Tokyo, Japan) equipped with a telephoto lens AF-S DX Micro Nikkor 85 mm (Nikin, Tokyo, Japan). Larvae were used 24–48 h after molting to the fifth instar. To prevent nonspecific biting due to starvation, all larvae fed on artificial diets for 2 minutes before use. For elucidation of the role of chemosensory organs, the MP, MG, or AN were quickly removed from a larva with fine forceps after the 2-minute feeding. Videos were taken for around 30 to 60 seconds after starting palpation of leaves and filter papers. Fresh host and nonhost leaves were collected within 2 weeks before use around the campuses of Tokyo University of Agriculture and Technology (Fuchu and Koganei, Tokyo, Japan) and stored at 4°C. For preparation of wiped leaves and leaf-surface extract, fresh leaves were gently wiped with Kimwipes (Nippon Paper Crecia Co., Tokyo, Japan) wetted with water or methanol. Kimwipes were dried in a drying machine, wetted with water or methanol again, and put into a centrifugal filter column (Merck Millipore, Darmstadt, Germany), and centrifuged to collect leaf-surface extracts. Filter paper (size: 1 cm long, 5 cm wide) was treated with surface extracts from a leaf. A larva moved freely after being put on a leaf, and all larvae showed palpation of all tested plant leaves. For the biting assay using commercial compounds, filter paper was impregnated with 100 μl of the stimulant solution. CGA (Nacalai tesque, Kyoto, Japan), ISQ (Extrasynthese, Lyon, France), Q3R (Extrasynthese), and/or βsito (Abcam, Cambridge, UK) were dissolved in ethanol, followed by vaporization of ethanol. Filter paper was inserted into a slit on foamed polystyrene (S1A Fig). A larva was put on the edge of the slit and videotaped. All larvae palpated on filter paper (S1B Fig). The fraction of larvae showing test or persistent biting was calculated for groups of 10 individuals.

### An agar-based food intake assay

To determine whether phytochemicals induce persistent biting, we used an agar-based food intake assay [22] with some modification. Agar-based diets basically contain 9% cellulose and 1% agarose in Elix water (Merck Millipore, Tokyo, Japan). Stimulants except for βsito and cellulose were mixed in advance and added to an agarose solution boiled using the microwave for dissolution. This mixture solution was vortexed well and poured into a 90-mm petri dish (Kanto chemical Co., Inc., Tokyo, Japan). After the agar-based diet solidified, βsito dissolved in EtOH was applied to the diet, followed by EtOH evaporation. For preparation of agar-based diets containing mulberry leaf extract, mulberry leaves were cut into small pieces and dried in a drying machine at 60°C. Dried leaves were crushed into powder. The composition of the mulberry leaf extract diet was 75% mulberry leaf powder (w/v), 1% agarose (w/v), and 24% Elix water (v/v) based on the ratio of dry and wet mass of intact mulberry leaves. Three to five larvae were put on each petri dish, and larval mass before and after 3 h feeding was measured. Increases of larval mass were regarded as amounts of food intake because no feces were observed after 3 h feeding at 25°C. We adopted increases of body mass (mg) as the index of food intake because no significant correlation between amount of food intake and the original body mass was observed (r = 0.11, $P > 0.5$) by Pearson's correlation analysis (S3A Fig).

## Tip recording

Tip recordings from all sensilla of the MP and the lateral and medial sensilla of the MG were conducted to determine whether neurons respond to CGA, ISQ, Q3R, and βsito. All 8 sensilla in the MP (S2A Fig) were stimulated together. Because LS and MS in MG have a *myo*-inositol neuron and a deterrent neuron that detects nicotine, respectively [17,30], they were used as positive control for stimulants of LS and MS (S2B and S2C Fig). Fifth-instar larvae starved until use were used for recording. CGA, ISQ, and Q3R were dissolved and serially diluted in water, whereas βsito was dissolved and diluted in ethanol. In dilution, solutions were stirred well for more than 30 minutes. We used 5 mM NaCl as an electrolytic solution. Methyl-β-cyclodextrin (βCD) was used as a complexing agent as described in Brown and colleagues [37] with some modifications. A solution of 2 mg/mL βsito in ethanol was added into 2.5% βCD in 5% NaCl solution so that the final concentration of βsito was 50 μM. The solution was put on a heating block at 80°C until it becomes clear. Tip recording methods were conducted based on Sasaki and colleagues [22]. Stimulants were filled up in glass microelectrodes and a silver wire inside of the electrode was connected to a TastePROBE amplifier (Syntech, Kirchzarten, Germany). The sensilla of MP and the sensilla of MG were capped with the recording electrode, using a micromanipulator to stimulate the chemosensory neurons and to record the response simultaneously. As a reference electrode, a fine silver wire insulated to its tip was inserted into the base of the maxilla. Electrical signals were recorded on a computer via a PowerLab 4/25 and analyzed using CHART 5 (ADInstrument, Bella Vista, Australia).

## Statistics

All statistical analyses were performed using Prism ver.8 (GraphPad, La Jolla, CA, USA). We used parametric tests (Fig 2A–2D, Fig 3B and 3C, and S1C Fig) and nonparametric tests (Fig 4A, 4B and 4C and S3C and S3D Fig). See S1 Data for more details (for example, exact sample sizes and *P*-value).

## Supporting information

**S1 Fig. Biting assay using filter paper.** (A) Biting assay using filter paper. Filter paper was inserted into a slit on foamed polystyrene. (B) All larvae palpated at the edge of the filter paper irrespective of treatment (left), and larvae showed test biting of treated filter paper. (C) Proportion of larvae (*n* = 10) showing test biting of filter paper treated with leaf-surface extracts of 2 edible leaves of *S. oleraceus* and *T. officinale* over 1 minute. Experiments were repeated as independent biological replicates (*n* = 3–5). Data are means ± SE. Statistical analysis was performed using one-way ANOVA followed by Tukey post hoc test. An asterisk indicates a significant difference (*$P$ < 0.05). For numerical raw data, please see S2 Data. (D) Representative raster plots of the timing and duration of biting behavior by larvae using filter paper treated with *M. alba* leaf-surface extract or a mixture of CGA, ISQ, and βsito. ANOVA, analysis of variance; CGA, chlorogenic acid; ISQ, isoquercitrin; βsito, β-sitosterol.
(TIF)

**S2 Fig. Tip recording of sensilla in the MG.** (A) Schematic of sensilla in the MP, which has 8 sensilla (5 putative gustatory and 3 olfactory sensilla). (B and C) Typical electrophysiological recordings from LS (B) and MS (C) in the MG in response to CGA, ISQ, Q3R, and βsito. Ino and Nico were used as positive controls for LS and MS, respectively. CGA, chlorogenic acid; Ino, *myo*-inositol; ISQ, isoquercitrin; LS, lateral styloconic sensillum; MG, maxillary galea; MP, maxillary palp; MS, medial styloconic sensillum; Nico, nicotine; Q3R, quercetin-3-*O*-rhamnoside; βsito, β-sitosterol.
(TIF)

**S3 Fig. Supplemental agar-based food-intake assay.** (A) Correlation between the original larval body mass and the increase in body mass after feeding a 100 mM Suc-containing agar-based diet for 3 h. Correlation coefficient (r) by Pearson correlation analysis ($n = 51$). (B) Representative raster plot of the timing and duration of biting when feeding agar containing 100 mM Suc. (C) Suc-dependent increase in larval weight in MP-ablated larvae after 3 h. (D) An effect of Glu on the increase in larval mass weight by agar-based food-intake assay. Magenta bars denote median. Statistical analysis was performed using Kruskal–Wallis test followed by Dunn test. "ns" indicates no significant difference; an asterisk indicates a significant difference (*$P < 0.05$; **$P < 0.01$). For numerical raw data, please see S2 Data. Glc, D-glucose; MP, maxillary palp; Suc, sucrose.
(TIF)

**S4 Fig. Effect of a mixture of inducers of test biting on increase in proportion of feeding larvae.** Sugars (10 mM sucrose, 5 mM *myo*-inositol, and 5 mM D-glucose) were added to the basic agar food (9% cellulose and 1% agar). The following inducers of test biting were added: 100 µM CGA, 1 µM ISQ, and 3 µg/cm$^2$ βsito. Data are from biological triplicate experiments ($n = 5$); error bars indicate SE. For numerical raw data, please see S2 Data. CGA, chlorogenic acid; ISQ, isoquercitrin; βsito, β-sitosterol.
(TIF)

**S1 Movie. Biting behavior of an intact larva towards a mulberry leaf.**
(MP4)

**S2 Movie. Biting behavior of an MP-ablated larva towards a mulberry leaf.** MP, maxillary palp.
(MP4)

**S3 Movie. Biting behavior of an MG-ablated larva towards a mulberry leaf.** MG, maxillary galea.
(MP4)

**S4 Movie. Biting behavior of an AN-ablated larva towards a mulberry leaf.** AN, antenna.
(MP4)

**S1 Data. Statistic information used in this study.**
(XLSX)

**S2 Data. Raw data used in this study.**
(XLSX)

## Acknowledgments

We thank Toru Shimada and Hiroki Takai (The University of Tokyo) for technical assistance in electrophysiological experiments at the early stage of this study, Naoaki Watanabe (Tokyo University of Agriculture and Technology) for assistance in sampling various plant leaves, Yutaka Banno (Kyushu University) for providing fresh *M. alba* leaves, and David G. Heckel (Max Planck Institute for Chemical Ecology) and Marian R. Goldsmith (University of Rhode Island) for helpful discussion and critical reading of the manuscript.

## Author Contributions

**Conceptualization:** Kana Tsuneto, Haruka Endo, Ryoichi Sato.

**Data curation:** Kana Tsuneto, Haruka Endo.

**Formal analysis:** Kana Tsuneto, Haruka Endo.

**Funding acquisition:** Haruka Endo, Ryoichi Sato.

**Investigation:** Kana Tsuneto, Haruka Endo, Fumika Shii.

**Methodology:** Kana Tsuneto, Ken Sasaki, Shinji Nagata.

**Project administration:** Haruka Endo, Ryoichi Sato.

**Resources:** Ken Sasaki, Shinji Nagata, Ryoichi Sato.

**Supervision:** Ken Sasaki, Shinji Nagata, Ryoichi Sato.

**Validation:** Kana Tsuneto, Haruka Endo.

**Visualization:** Kana Tsuneto, Haruka Endo, Fumika Shii.

**Writing – original draft:** Kana Tsuneto, Haruka Endo.

**Writing – review & editing:** Kana Tsuneto, Haruka Endo, Ken Sasaki, Shinji Nagata, Ryoichi Sato.

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
