## [Editor Report · Decision Letter 0]

23 Jan 2020

Dear Dr Endo, 

Thank you for submitting your manuscript entitled "Diet choice: The two-factor authentication system of silkworm larvae" for consideration as a Research Article by PLOS Biology.

Your manuscript has now been evaluated by the PLOS Biology editorial staff as well as by an academic editor with relevant expertise and I am writing to let you know that we would like to send your submission out for external peer review. Please note, however, that the outcome of our discussion of your manuscript is that we have some reservations as to the overall strength of novel biological insight offered by your data. We will need to be persuaded by the reviewers that the paper has the potential after revision to offer the significant advance that we require for publication.

Before we can send your manuscript to reviewers, we need you to complete your submission by providing the metadata that is required for full assessment. To this end, please login to Editorial Manager where you will find the paper in the 'Submissions Needing Revisions' folder on your homepage. Please click 'Revise Submission' from the Action Links and complete all additional questions in the submission questionnaire.

Please re-submit your manuscript within two working days, i.e. by Jan 25 2020 11:59PM.

Kind regards,

Lauren A Richardson, Ph.D

Senior Editor

PLOS Biology

---

## [Decision Letter · Decision Letter 1]

8 Mar 2020

Dear Dr Endo,

Thank you very much for submitting your manuscript "Diet choice: The two-factor authentication system of silkworm larvae" for consideration as a Research Article at PLOS Biology. Your manuscript has been evaluated by the PLOS Biology editors, an Academic Editor with relevant expertise, and by three independent reviewers.

As you will see, the reviewers find the conclusions interesting and valuable for the field, but they also raise several issues that need to be addressed to strengthen the results and conclusions. After discussing the reviews with the Academic Editor, we feel that the inclusion of the additional experiments suggested by the reviewers could improve the manuscript to a sufficient level for us to consider it for publication.

In light of the reviews (attached below), we will not be able to accept the current version of the manuscript, but we would welcome re-submission of a much-revised version that takes into account the reviewers' comments. We cannot make any decision about publication until we have seen the revised manuscript and your response to the reviewers' comments. Your revised manuscript is also likely to be sent for further evaluation by the reviewers.

We expect to receive your revised manuscript within 2 months. 

**IMPORTANT - SUBMITTING YOUR REVISION**

*Re-submission Checklist*

*Published Peer Review*

*PLOS Data Policy*

*Blot and Gel Data Policy*

Sincerely,

Ines

--

Ines Alvarez-Garcia, PhD

Senior Editor

PLOS Biology

Carlyle House, Carlyle Road

Cambridge, CB4 3DN

+44 1223–442810

Reviewers’ comments

Rev. 1:

This manuscript reports an interesting study of neurophysiological and behavioral mechanisms of host-plant acceptance by larval silkworms. As far as I know, the identification of the relative roles of the maxillary palps and galeae in the process of host acceptance represents a new finding. Previous work had mostly focused on particular subsets of the taste system, such as the medial or lateral sensilla of the galeae, or behavioral mechanisms by themselves. And limited work had been done on taste receptors of the maxillary palps. In the present manuscript, the coupling of the electrophysiological work on the taste system encompassing the palps and galeae with behavioral assays was especially refreshing and insightful. The authors conclude that the silkworm accepts a host plant via a "two-factor authentication system" in that stimulation of the taste cells of the maxillary palps first cause biting of the plant, and, second, stimulation of the taste cells of the galeae are responsible for sustained biting (i.e. feeding). This finding has the potential to explain host acceptance behavior by the larvae of other species of Lepidoptera and perhaps other insect orders.

Despite my enthusiasm, I have several general criticisms that I introduce here and pursue in greater detail in my specific comments. First, the interpretation of the neurophysiological mechanism of the first step of host acceptance is missing the potential role of olfaction during palpation. The design of the experiments could not determine this possibility in some cases because olfactory responses were not measured. In one other experiment in which the antennae were ablated, there is evidence that this treatment increases biting (Fig. 1), which is consistent with the hypothesis that olfactory stimuli play a role in the first step of host acceptance by deterring biting. Second, there are some significant gaps in scholarship, particularly from the late 1990's through the early 2000's. Many papers were published elucidating gustatory neurophysiological mechanisms of host-plant acceptance by caterpillars of several species during this time. These uncited papers are important in placing the findings of this manuscript in context. For example, Chapman 2003 (see references below) proposed a generalized gustatory neurophysiological mechanism of host-plant acceptance by phytophagous insects (especially caterpillars). The present manuscript would greatly benefit from placing its content in that context. The authors should discuss whether their findings agree with and extend the Chapman proposal in new ways or whether they argue against the Chapman proposal? I will admit that this field is not my main field, so I don't know much of the literature since the Chapman 2003 review. But I see that the review has been cited 192 times as of today, so there may be much more recent work to cite here.

Specific comments

Abstract, line 15. Replace "nutritional value" with "quality as food" because the food quality of a plant represents both nutritional value and effects of plant secondary metabolites. On the same line, replace "by which insects select host plants through sensing a variety of gustatory inputs" with "of host-plant selection" to avoid lengthy redundancy.

Abstract, line 17. I strongly suggest replacing "authentication" with "host recognition" system. The latter is a conventional phrase that many researchers would understand. Moreover, the authors use "host recognition" elsewhere in the manuscript. There is no need to introduce a new term here. This change should be made to the title and all other parts of the manuscript.

Abstract, lines 22-24. This description of the electrophysiological role of the maxillary galea is lacking much understanding about this part of the taste system, as described in various other studies and summarized in Chapman 2003. Sucrose is a critical stimulant, but secondary metabolites are key as well, with either stimulating or deterrent effects on biting and feeding.

Introduction, line 30. Add "host" before "plants" for clarity, and remove "and carving their own niche" because it does not add any real information.

Introduction, lines 35-37. An important reference that is missing here is a book on this topic by Chapman and de Boer from 1995 (see references).

Introduction, lines 41 and 42. Add a hyphen, so it reads "host-plant selection." Elsewhere, this version of the phrase is used.

Introduction, lines 41-42. I would like to see a bit more of a summary of what's known and what isn't here. It's not entirely fair to say "the neuronal basis" of host-plant selection "and its roles at the leaf surface in whole host-plant selection or leaf choice are poorly understood." While there is certainly more understanding needed, this characterization of the state of knowledge is rather dismissive of what is known (some of which is described in the following paragraph).

Introduction, line 45. Here is where the Chapman 2003 review should be cited. This paragraph should be expanded with some of the information in that paper and related ones.

Introduction, line 49. There is a grammatical issue with "number of feces." I'm not sure what the authors are trying to say here, so I can't really offer a correction.

Introduction, lines 50-51. The end of this paragraph places this study in a very narrow context, as if everyone in the field was mostly interested in understanding host-plant selection by Bombx mori. In fact, B. mori is one of several model species used in this field, and the highly relevant work on these other species can help place the B. mori work in a broader context.

Introduction, lines 57-58. Replace "authentication" with "host recognition" and replace "that underlies oligophagy" with "responsible for the initiation of feeding" for clarity.

Results, line 62. Replace "behavior towards" with "of" for simplicity and clarity.

Results, line 78. Replace "whether" with "how" and replace "contributes" with "contribute" for grammatical improvement.

Results, line 81. The introduction of Lactuca indica here is confusing because this plant is not tested in the study reported here. I wondered if this name was a mistake because it seems like the authors might be referring to Sonchus oleraceus instead. Also in line 81, replace "consumed a relatively small amount" with "consumed in relatively small amounts."

Results, lines 101-102. Again, Lactuca indica is mentioned here, but not elsewhere.

Results, line 111. Replace "were" with "was."

Results, lines 118-119. Here is where I think the possible role of olfactory inputs should be mentioned.

Results, lines 123-124. Here is where I think the broader range of stimulants should be considered for taste cells of the lateral sensilla. It might be the case that LS taste cells only respond to these sugars, but that is not the case for other caterpillar species.

Results, line 135. The "important role of the MG in modulating the amount of food intake" is already known from work on other species. If this information were added to the Introduction section, the statement here could say that this finding confirms "the important role…" rather than suggesting that this is a new finding.

Results, line 137. Replace "in consistent" with "in agreement."

Results, line 140. Remove "that" after "whether."

Discussion, lines 150-151. I suggest changing "comprehensively upgrade" to something like "provide new."

Discussion, line 153. Here, as in the Introduction, the state of knowledge of this field is mischaracterized. Adding the new scholarship should help the authors re-characterize the previous state of knowledge, so their contribution can be more accurately described.

Discussion, lines 157-161. Here is another place where I think it is important to consider the possible role of olfactory inputs in modulating the biting response. Note that the agar-based diet is not likely to have the olfactory stimuli that leaves would have.

Discussion, lines 164-167. I don't find this argument very convincing. In natural forests, canopy branches and leaves of multiple tree species often intersect, and individual caterpillars sometimes fall or walk off their host plant. Even dietary specialist caterpillars do not necessarily stay on the same plant for their entire larval lives. Therefore, it is likely to be important for dietary specialists of trees to have the ability to recognize their specific hosts. I agree that such specific signals do not necessarily have to be single compounds that are unique to the host, even though there are cases in which this is true (Bernays et al. 2003, see references). As argued at the end of the following paragraph, unique combinations of compounds can give specific signals, as seen in pheromone-based species recognition systems of insects.

Discussion, lines 169-178. This paragraph is missing much recent scholarship (see references below) on caterpillar neurophysiological mechanisms of feeding.

Discussion, lines 183-184. Replace "taste modalities such as sweet, bitter…" with "chemical tastants wuch as sugars, amino acids, secondary metabolites…" The former are subjective sensations, whereas my suggestion gives objective information.

Discussion, line 191. Replace "The MG has been believed to play a main role in…" with "The MG has been shown to play a main role in…" This would be a good place to cite Chapman 2003.

Discussion, line 193. Note that the MG in at least one species of caterpillar is known to respond to a stimulating secondary metabolite (pyrrolizidine alkaloids) at 10-12 M. Also replace "authentication" with "host recognition."

Discussion, line 198-199. It's also possible that olfaction plays a role in deterrent inputs provided by the MP, which contains gustatory and olfactory sensilla.

Discussion, lines 199-200. Replace "authentication" with "host recognition" and replace "negative selection" and "positive selection" with terms that are not conventionally understood to have different meanings in biology (evolutionary biology in this case). In this context, I suggest stimulating and deterrent effects on feeding.

Discussion, lines 202-206. This final paragraph reads as if the authors are struggling to place their work in a larger context. I think they will find it easier to do this when they re-frame the Introduction with additional scholarship (see references). The revised Introduction will help set up their study and the final points in the Discussion section can more clearly specify the authors' contribution in addition to specific gaps in the knowledge that future research can address.

Replace "authentication" in the first sentence. Later, the statement "oligophagy and polyphagy are likely consequences of the strict and loose restriction, respectively, of feeding initiation by the MP" confuses proximate and ultimate mechanisms in biology. It is almost certainly the case that the characteristics of the MP and their proximate effects on feeding behavior are a consequence of natural selection for narrow or broad diets in herbivorous insects. I suggest either rewriting this phrase or omitting it. Finally, the concluding line in the paragraph is not truly informative or helpful. At worst, it misleads the reader to believe that "mechanisms underpinning host-plant selection" are not really understood at all. In fact, there are entire books written about this topic (e.g., Bernays and Chapman, 1994, Host-plant selection by phytophagous insects).

Methods, line 215. Change to "Videos of silkworm feeding on leaves and filter paper…" The videos are great, by the way.

Methods, line 218. Add "on" after "fed."

Methods, line 219. Add "the" before "maxillary" and make palp, galea, and antenna plural in this sentence. Also, replace "was" with "were."

Methods, line 221. Replace "towards" with "of."

Methods, line 223. Replace "in" with "at."

Methods, line 225. Replace "wet" with "wetted."

Methods, line 228. The phrase "all larvae started finally palpation towards all kinds of plant leaves" is not grammatically correct or clear. I don't know exactly what the authors are trying to say, so I don't have any correction to offer. On this line at the end, add "the" after "For."

Methods, line 229. Add "of the" before "stimulant."

Methods, line 234. Replace "investigated" with "calculated."

Methods, line 236. Add "an" before "agar-based."

Methods, line 241. Add "the" before "agar-based."

Methods, line 243. Provide the temperature at which the leaves were dried in the oven.

Methods, line 245. Add "the" before "ratio" and replace "weight" with "mass" here and throughout this paragraph.

Methods, line 252. Add "the" before "lateral" and change "sensillum" to "sensilla."

Methods, line 256. Starving the larvae here and in the food intake assay can affect the results of these experiments. In addition to generalized effects of food deprivation, there can be specific effects on taste and feeding responses that reflect the insect's physiological state of nutrient deprivation. For example, dietary deficiencies in carbohydrates tend to increase the taste and feeding responses of caterpillars to carbohydrates (e.g., Bernays et al. 2004a in references). I doubt this issue has large effects on the results reported here, but I want the authors to be aware of this phenomenon in case they are not already.

Methods, line 257. The part of the line after "whereas" is incomplete.

Methods, line 261. Replace "being" with "it became."

Figure 1 legend, line 379. Change "persistent bites." to "persistent bites, respectively."

Figure 4. Replace "weight" in the y-axis labels of panels A, B, C with "mass." Same for the axis labels in Fig. S3.

Figure 4 legend, line 409. Replace "weight" with "mass." Same for the legend of Fig. S3.

References

Bernays and Chapman. 1994. Host-plant selection by phytophagous insects. Chapman & Hall.

Chapman and de Boer, editors. 1995. Regulatory mechanisms in insect feeding. Chapman & Hall.

Bernays et al. 1998. Plant acids modulate chemosensory responses in Manduca sexta larvae. Physiological Entomology 23:193-201.

Bernays and Chapman. 2001. Taste cell responses in the polyphagous arctiid Grammia geneura: towards a general pattern for caterpillars. Journal of Insect Physiology 47:1029-1043.

Bernays et al. 2002. A highly sensitive taste cell for pyrrolizidine alkaloids in the lateral galeal sensillum of a polyphagous caterpillar, Estigmene acrea. Journal of Comparative Physiology A 188:715-723.

Bernays et al. 2002. A taste receptor neurone dedicated to the perception of pyrrolizidine alkaloids in the medial galeal sensillum of two polyphagous caterpillars. Physiological Entomology 27:312-321.

Bernays et al. 2003. Taste receptors for pyrrolizidine alkaloids in a monophagous caterpillar. Journal of Chemical Ecology 29:1709-1722.

Chapman. 2003. Contact chemoreception in feeding by phytophagous insects. Annual Review of Entomology 48:455-484.

Bernays et al. 2004a. Changes in taste receptor cell sensitivity in a polyphagous caterpillar reflect carbohydrate but not protein imbalance. Journal of Comparative Physiology A 190:39-48.

Bernays et al. 2004b. Gustatory responsiveness to pyrrolizidine alkaloids in the Senecio specialist, Tyria jacobaeae (Lepidoptera, Arctiidae). Physiological Entomology 29:67-72.

Rev. 2:

In this manuscript, the authors investigated the diet choice behavior of the silkworm, a famous specialist which feeds on mulberry and several close species, with different bioassay methods. They explained how mulberry preference of the silkworm was related to the appropriate combination of the three compounds in mulberry leaves. They further proposed that a two-factor authentication system, controlled by two peripheral gustatory organs of maxillary palp (MP) and maxillary galea (MG), and driven by six phytochemicals, that underlies oligophagy in silkworms. This work is interesting and provides valuable information in diet choice of phytophagous insect species. However, the conclusion is too rough to be convincing.

Assuming that combination of three compounds are responsible for mulberry preference, how to explain that silkworm larvae also feed on artificial diet (should be different compound combination even this diet contained mulberry leaves)? It was reported that polyphagous silkworm mutant strains feed on non-mulberry diets (Iizuka et al., 2012). A recent study also showed that a gustatory receptor gene determines the silkworm feeding preference (Zhang et al., 2019). In addition, possibly there are other phytochemicals in mulberry leave will also induce feeding behavior including palpation and biting. The reviewer strongly suggests that using a polyphagous mutant strain, such as Sawa-J, to make clear comparison.

The tip recording results showed in Figure 3A and Figure S2B and C were very confusing. The concentration of chlorogenic acid (CGA), isoquercitrin (ISQ), quercetin-3-O-rhamnoside (Q3R), and β-sitosterol used in MP and MG testing were quite different. In addition, stimulus intensity is usually a positive function between stimulus concentration and the number of spikes elicited in a peripheral neuron. Thus, the authors should add the experiment of tip recording (both MP and MG) with gradient concentration. Otherwise, the present results could not support the conclusion that MP responded to the four compounds and MG did not respond to these compounds.

Rev. 3: Heiko Vogel - note that this reviewer has waived anonymity

This is a very nice manuscript by Tsuneto and colleagues, who have studied the perception system by which silkworm larvae decide to continuously feed (or reject) specific host plant species/plant diets.

The focus of the manuscript is on a proposed "two-factor" authentication system which larvae utilize to first perform leaf palpation, using the maxillary palp, and detecting low levels of leaf surface compounds. This first step is followed by test bites of the larvae, and includes subsequent detection of leaf sap compounds by chemosensory neurons localized on the maxillary galea and finally the induction of persistent feeding - in case the larvae encounter the correct host plant.

The authors start by illustrating what is known from the literature on decision making related to herbivorous insect feeding. Prior work from the 1960s and 1970s and more recent papers from several authors have indicated leaf surface compounds as important cues for herbivorous insects to accept host plants. In addition, numerous compounds (also mostly present on leaf surfaces) have been identified which induce oviposition of female insects. However, the major factors, insect physiology and molecular players in host plant acceptance and more generally the "turning points" in decision making of herbivorous larvae when feeding on plants were not well understood.

I found it especially interesting to see that the larval "test biting" clearly shows a frequency gradient on leaves of different host plants. In contrast to this, the persistent biting falls into two clearly separated groups: one group with the acceptable host plants show a gradient of larvae performing persistent biting and the other group (inedible leaves) display essentially zero persistent biting of larvae.

To summarize, this work does a nice job of combining multiple types of experimental approaches/assays. It is a nice finding that shows how a specialized (oligophagous) herbivorous insect can - by going through a series of "decision points" - ultimately discriminate host and non-host plants and thus accept or reject a specific plant diet. The study is well conceived and connects well with previous results, and the methodology employed in the different aspects is state of the art. The results are properly documented, and their interpretation in the discussion is quite reasonable.

I have just minor suggestions for improvement, but see no major flaws that can't be easily addressed with some additions/modifications.

Minor comments:

In all of the figures it would be helpful to provide explanations for all of the abbreviations used. Otherwise the interpretation of at least parts of the figures is severely hampered.

In the discussion the authors compare their findings on compound perception, neuronal perception and decision making in larvae to adult oviposition on plant leaves. Although superficially touched upon, here or in other parts of the discussion the authors could add a few sentences on what their findings might mean in general for monophagous/oligophagous herbivores (i.e. specialist) versus highly polyphagous (i.e. generalist) ones. Do polyphagous species also require a "two-factor authentication system" - and if so what would be the main discriminatory chemical compounds. Or are polyphagous species instead rather relying on deterrents for decision making and host plant acceptance. Even though this would be rather speculative, it would provide a better approach at the more general question of what defines a oligophagous versus a highly polyphagous herbivore.

---

## [Decision Letter · Decision Letter 2]

26 Jun 2020

Dear Dr Endo,

Thank you for submitting your revised Research Article entitled "Diet choice: The two-factor host acceptance system of silkworm larvae" for publication in PLOS Biology. I have now obtained advice from two of the original reviewers and have discussed their comments with the Academic Editor. 

Based on the reviews, we will probably accept this manuscript for publication, assuming that you will modify the manuscript to clarify some points in the text raised by Reviewer 1 (see file attached to this email). Please also make sure to address the data and other policy-related requests noted at the end of this email.

We expect to receive your revised manuscript within two weeks. Your revisions should address the specific points made by each reviewer. In addition to the remaining revisions and before we will be able to formally accept your manuscript and consider it "in press", we also need to ensure that your article conforms to our guidelines. A member of our team will be in touch shortly with a set of requests. As we can't proceed until these requirements are met, your swift response will help prevent delays to publication.

*Copyediting*

*Published Peer Review History*

*Early Version*

*Submitting Your Revision*

Sincerely,

Ines

--

Ines Alvarez-Garcia, PhD

Senior Editor

PLOS Biology

Carlyle House, Carlyle Road

Cambridge, CB4 3DN

+44 1223–442810

DATA POLICY: PLEASE READ

Many thanks for sending us the raw data underlying all the graphs shown in the figures. I have a couple of requests that remain to be addressed:

- Please delete the tab labelled Fig. 3D from the Data S2 Raw Data file – there is no D section in this figure and the data seems to be the same than Fig. 3C.

- Please also ensure that figure legends in your manuscript include information on WHERE THE UNDERLYING DATA CAN BE FOUND.

Reviewers' comments

Rev. 1:

This manuscript is much improved, thanks to the careful attention of the authors to the comments by the reviewers. My minor comments were addressed to my satisfaction in this revision. My major comments were mostly addressed to my satisfaction, even if I disagree with one or two responses from the authors. 

For example, I am not convinced by the argument that "silkworm larvae have no reasons to deter biting of mulberry leaves" as rationale for discounting the possibility that olfactory inputs might deter biting. From an evolutionary ecology perspective, I can think of at least one reason that a caterpillar might use olfactory cues from its host plant to decide whether to feed or move elsewhere to feed. I preface my speculation with the important point that the decision to feed by a dietary specialist herbivore is not merely an automatic response to the correct host plant species (as the authors suggest in this manuscript). Insect herbivores use chemical cues to perceive phenotypic variation among individual host plants and parts of host plants, and they can respond adaptively to such variation (e.g., Knolhoff and Heckel 2014, Annual Review of Entomology 59:263). Even dietary specialist herbivores do not respond to every plant of the correct host species in the same way. For example, they might discriminate among plants with high or low resistance traits, some of which are inducible. Regarding olfactory cues, plant volatiles induced by herbivores contain information that predators and parasitoids use to locate their herbivorous prey and hosts (e.g., Heil 2014, New Phytologist 204:297), and herbivores can also use these volatile compounds emitted by plants to accept or reject host plants (e.g., Kessler and Baldwin 2001, Science 291:2141; De Moraes et al. 2001, Nature 410:577). Granted, demonstrations of this phenomenon are limited to adult moths and oviposition behavior. However, caterpillars have been shown to respond behaviorally to the induced chemistry of their host plants over very short time and spatial scales (e.g., Perkins et al. 2013, Proceedings of the Royal Society of London B 280:20122646). Even though Bombyx mori is a domesticated species, it might retain some ability to use olfactory cues from its host plant in ways that would have been adaptive for its wild ancestors.

I understand that my hypothesis for this unexplained detail in Fig. 1C is highly speculative, thus I am not asking for the authors discuss it. However, I want them to consider it for future work on the functional role of olfaction in caterpillar feeding behavior.

The only revisions I would insist on at this point are some minor things to correct grammatical mistakes in sections that were re-written or added. My corrections and edits for these points are given as sticky notes on the pdf version of the manuscript attached with my review.

Rev. 2: 

Authors made significant improvement in the revised manuscript and may be accepted now.

---

## [Editor Report · Decision Letter 3]

17 Aug 2020

Dear Dr Endo,

On behalf of my colleagues and the Academic Editor, Anurag A Agrawal, I am pleased to inform you that we will be delighted to publish your Research Article in PLOS Biology. 

Early Version

PRESS 

Kind regards,

Alice Musson

Publishing Editor, 

PLOS Biology

on behalf of

Ines Alvarez-Garcia,

Senior Editor

PLOS Biology